# Genome-Wide Expression Profiling Analysis of Kiwifruit *GolS* and *RFS* Genes and Identification of *AcRFS4* Function in Raffinose Accumulation

**DOI:** 10.3390/ijms23168836

**Published:** 2022-08-09

**Authors:** Jun Yang, Chengcheng Ling, Yunyan Liu, Huamin Zhang, Quaid Hussain, Shiheng Lyu, Songhu Wang, Yongsheng Liu

**Affiliations:** 1College of Horticulture, Anhui Agriculture University, Hefei 350002, China; 2State Key Laboratory of Subtropical Silviculture, Zhejiang A&F University, 666 Wusu Street, Hangzhou 311300, China

**Keywords:** kiwifruit, galactinol synthase, raffinose synthetase, gene family, abiotic and biotic stresses

## Abstract

The raffinose synthetase (*RFS*) and galactinol synthase (*GolS*) are two critical enzymes for raffinose biosynthesis, which play an important role in modulating plant growth and in response to a variety of biotic or abiotic stresses. Here, we comprehensively analyzed the *RFS* and *GolS* gene families and their involvement in abiotic and biotic stresses responses at the genome-wide scale in kiwifruit. A total of 22 *GolS* and 24 *RFS* genes were identified in *Actinidia chinensis* and *Actinidia eriantha* genomes. Phylogenetic analysis showed that the *GolS* and *RFS* genes were clustered into four and six groups, respectively. Transcriptomic analysis revealed that abiotic stresses strongly induced some crucial genes members including *AcGolS1/2/4/8* and *AcRFS2/4/8/11* and their expression levels were further confirmed by qRT-PCR. The GUS staining of *AcRFS4Pro*::GUS transgenic plants revealed that the transcriptionlevel of *AcRFS4* was significantly increased by salt stress. Overexpression of *AcRFS4* in *Arabidopsis* demonstrated that this gene enhanced the raffinose accumulation and the tolerance to salt stress. The co-expression networks analysis of hub transcription factors targeting key *AcRFS4* genes indicated that there was a strong correlation between *AcNAC30* and *AcRFS4* expression under salt stress. Furthermore, the yeast one-hybrid assays showed that *AcNAC30* could bind the *AcRFS4* promoter directly. These results may provide insights into the evolutionary and functional mechanisms of *GolS* and *RFS* genes in kiwifruit.

## 1. Introduction

The productivity, development and growth of plants can be seriously affected by diverse environmental stresses, such as low or high temperatures, drought, salinity, pests and diseases [1]. Under stress conditions, plants perform a series of physiological and biochemical responses, cellular and molecular mechanisms by activating many stress-responsive genes or transcription factors and synthesizing various functional proteins [2]. At the cellular level, plants produce many compatible molecules solutes, such as mannitol, proline and oligosaccharides (galactinol, trehalose, raffinose and stachyose), which serve as regulatory compounds to deal with these abiotic stresses [3]. Raffinose family of oligosaccharides (RFOs), especially raffinose, plays an essential role in increasing the osmotic pressure in cells. Furthermore, raffinose also acts as antioxidants, signals, transport and storage of carbon, and membrane stabilizer to protect the plant cell from dehydration [4].

It is well known that raffinose and galactinol are two major oligosaccharides. Galactinol synthase (*GolS*; EC: 2.4.1.123) is responsible for raffinose biosynthesis in the initial stage, which is catalyzed by the reaction of Myo-inositol and UDP-galactose to generate galactinol [5]. Galactinol synthase is an important enzyme regulating the osmoprotectant function of RFOs in plants [6]. It has been reported that *CsGolS1* was up-regulated by abiotic stresses and *CsGolS2* were up-regulated by biotic stress in *Camellia sinensis* [7]. Recently, Cui et al. reported that over-expression of *DaGolS2* in rice increased the tolerance to drought and cold stresses by inducing the accumulation of raffinose and decreasing ROS levels [8]. Regarding wheat, *TaGolS1*, *TaGolS2* and *TaGolS3* have been well clarified. Over-expression of *TaGolS1* or *TaGolS2* in rice was found to accumulate the content of galactinol and raffinose, then improved cold stress tolerance [9]. The mRNA level of *AtGolS1* and *AtGolS2* in *Arabidopsis* were expressed under drought and salinity stresses, while *AtGolS3* responded to cold [10,11]. Higher galactinol and raffinose contents were exhibited, and increased tolerance to water deficit in the apple *MdGolS2* transgenic *Arabidopsis* plants, compared with the wild type [12]. To date, the *GolS* genes have been identified in *Nicotiana tabacum* and *Brassica napus* [13], *Zea mays* [14], *Manihot esculenta*
*Crantz* [15], *Solanum lycopersicum* and *Brachypodium distachyum* [16], and *Sesamum indicum* [17] at the genome-wide level. Raffinose synthase (*RFS*, EC 2.4.1.82) another key enzyme for raffinose synthesis, also plays a crucial role in response to abiotic and biotic stresses [4]. The *RFS* genes were isolated from *Pisum sativum* [18], *Gossypium* [19], *Cucumis sativus* [20], *Oryza sativa* [21] and *Arabidopsis thaliana* [22]. *PtrERF108* played a positive role in cold tolerance by modulation of raffinose synthesis via regulating *PtrRafS* in *Poncirus trifoliata* [23]. The latest report indicated that the *Vitis vinifera VviRafS5* was up-regulated by cold and ABA but not by heat and salt stresses. Overexpression of *VviRafS5* in yeast, can significantly increase the content of raffinose [24]. To the best of our knowledge, neither *GolS* nor *RFS* gene has been identified in kiwifruit.

Kiwifruit belongs to *Actinidia*, which contains 55 species and approximately 75 taxa originated from China [25]. It is commonly called “the king of fruits” as its remarkably high nutritional value including vitamin C and minerals [26]. In recent years, more and more varieties of kiwifruit were commercialized. Notably, the yield and quality of kiwifruit can be impaired by abiotic and biotic stresses, including drought, salinity, low or high temperatures and bacterial canker [27,28,29,30,31]. So far, four kiwifruit genomes have been sequenced, one belonging to *A. eriantha* and the rest belonging to *A. chinensis* [32,33,34,35]. There is still no comprehensive study of the evolution and expression patterns of the *RFS* and *GolS* gene family in kiwifruit. Gene structure, phylogenetic analyses, chromosomal location and conserved motifs were performed in the present study. In addition, we also analyzed transcript profiles of the *AcGolS* and *AcRFS* gene family in different tissues and under abiotic and biotic stress. Key *AcRFS* and *AcGolS* genes with high expression levels under salt, cold and bacterial canker stress were selected to construct co-expression networks with all transcription factors in kiwifruit. Finally, the biological functions of *AcRFS4* under salt stress were also investigated by stable transformation of *Arabidopsis*. The results suggested that *AcRFS4* might enhance salt tolerance in plants by modulation of raffinose content. 

## 2. Results

### 2.1. Identification, Characterization, and Phylogenetic Analysis of the GolS and RFS Genes in Kiwifruit 

Using seven *AtGolS* proteins from *A. thaliana* as query sequences, Twenty-two *GolS* genes in kiwifruit were identified (Appendix A). We named *AcGolS1*-*AcGolS9* in *A. chinensis* and *AeGolS1*-*AeGolS13* in *A. eriantha* based on the gene names in *A. thaliana*. Twenty-four *RFS* genes were also identified in the kiwifruit genome, and designated as *AcRFS1*-*AcRFS12* in *A. chinensis* and *AeRFS1*-*AeRFS12* in *A. eriantha* (Appendix A). The characterization information of the *GolS* and *RFS* genes in kiwifruit, including locus ID, linkage group distribution, the length of coding sequences, molecular weight (MW), theoretical isoelectric point (pI), and subcellular localization prediction were listed in Appendix A. 

Phylogenetic trees based on four species were created to describe the evolutionary of the *RFS* and *GolS* gene families. As shown in Figure 1A, four and two *GolS* genes from *A. eriantha* and *A. chinensis*, respectively, were found in GolS-I. In the clade GolS-II, only *AeGolS6* was found. Six members of GolS-III were found, including three *GolS* genes from *A. chinensis* and *A. eriantha*, respectively. GolS-VI was the largest clade, with nine members, including five and four *GolS* genes of *A. chinensis* and *A. eriantha*, respectively. The result showed that the *RFS* genes were classified into six groups (RFS-I to RFS-VI) (Figure 1B). Like *GolS* genes, *RFS* genes in kiwifruit have a closer relationship with *Camellia sinensis*. In the clade RFS-IV, all *RFS1* genes were found from *C. sinensis*, *A. thaliana, A. eriantha* and *A. chinensis* (Figure 1B). 

### 2.2. Structure, Protein Motif, and Cis-Element in Promoter Regions of the GolS and RFS Genes

To investigate the exon-intron organization of *AcGolS* and *AeGolS* genes, five and four introns were observed in most *GolS* genes in kiwifruit (Figure 2A). We saw only one exon in *AcGolS6* and *AeGolS13* (Figure 2A). *AeGolS5* contained the five introns, which had the most significant introns in *GolS* genes from *A. eriantha* (Figure 2A). Many *GolS* genes in the same groups had similar exon-intron structure, which was highly conservative in kiwifruit (Figure 2A). The numbers of *RFS* introns varied from 3 to 14 (Figure 2B). The *RFS* genes belonging to RFS-III and RFS-IV had more introns than other (Figure 2B). *AeRFS5* contained 14 introns in RFS-III, while other genes contained 13 exons (Figure 2B). The distribution of exons indicated that *RFS* genes belonging to the same group had similar gene structures and exon numbers (Figure 2A,B). A total of 10 motifs named motif1 to motif10 were predicted in the kiwifruit *RFS* and *GolS* genes. Except for *AeGolS5/6/7/8*, all *GolS* genes contained motif2 (Figure 2A). Two motifs were found in *AcGolS6* and *AcGolS8* (Figure 2A). Motif3 was distributed on GolS-II-IV (Figure 2A). All motifs were observed in *AeRFS1/4/6/10* (Figure 2B). There were more than seven motifs in most *RFS* genes in kiwifruit (Figure 2B). The 2000 bp sequences upstream of the start codon of *AcGolS* and *AcRFS* genes were used to analyze cis-elements in their promoter regions by PlantCARE. Sixteen putative cis-elements responsive to biotic stresses [including W-box, TC-rich repeats (defense and stress-responsive element), and WUN-motif (wound-responsive element), LTR (low-temperature responsive element), MBS (MYB binding site), and phytohormones such as ABA [abscisic acid, ABRE (ABA-responsive element)], auxin (TGA-element, AuxRR-core), GA (gibberellin, GARE), JA (jasmonic acid, CGTCA-motif), SA (salicylic acid, TCA-element) and endosperm development (P-box) were present in the promoters of *GolS* and *RFS* genes in kiwifruit (Appendix A). The same conserved motifs in homologous *RFS* genes might have similar functions and correlations between evolutionary relationships and conserved motifs.

### 2.3. Duplication, Synteny, and Chromosomal Distribution of GolS and RFS Genes

The segmental and whole genome duplications of *GolS* and *RFS* genes were determined in *A. chinensis* and *A. eriantha*. A total of 18 pairs of *GolS* genes segmental duplicates were observed in the kiwifruit genome (Figure 3A). All *RFS* genes within the kiwifruit genome had segmental duplicates, suggesting a high level of conservation of the *RFS* gene family (Figure 3B). A comparative syntenic map of *A. chinensis*, *A. eriantha* and *A. thaliana* was constructed (Appendix A). Interestingly, we found that 6 *AcGol* and 9 *AeGolS* genes had a syntenic relationship with *AtGolS* genes and located on chr1 (Appendix A). 7 *AcRFS* and 8 *AeRFS* genes had a syntenic relationship with *AtRFS* genes and located on chr1, chr3, chr4 and chr5 (Appendix A). All *GolS* and *RFS* in kiwifruit were mapped to the 37 chromosomes. The chromosomal distribution information of *GolS* and *RFS* genes was shown in Appendix A. 

### 2.4. Expression Patterns Analysis of AcGolS and AcRFS Genes in Different Plant Tissues, during the Fruit Development and under Hormone-Induced Conditions

As shown in Figure 4A, six of nine *AcGolS* g enes (except *AcGolS1/3/5*) were highly expressed in the leaf. *AcGolS3* expression was higher in the stem than in other tissues. *AcGolS1* was found to be highly expressed in the shoot. *AcGolS2/5/8/9* were mildly expressed in the shoot (Figure 4A). Only three *AcGolS* genes (*AcGolS3/5/8*) had low expression during the fruit development. Six of nine *AcGolS* genes were induced by abscisic acid (Figure 4A). *AcGolS1/2/5/7* were induced in response to GA (gibberellins). Only *AcGolS9* was strongly expressed under salicylic acid (SA) treatments (Figure 4A).

*AcRFS8/9/10* exhibited specific high expression in stem (Figure 4B). *AcRFS3/4/12* exhibited relatively low expression levels in all tissue (Figure 4B). *AcRFS1/2/11* were strongly expressed in the flower. Only *AcGolS9* was strongly expressed under salicylic acid treatments (Figure 4B). Among *AcRFS* genes, only *AcRFS5* had a high expression in the shoot (Figure 4B). Six *AcRFS* genes (except *AcRFS*1/3/5) were highly expressed in the leaf (Figure 4B). Five *AcRFS* genes (*AcRFS2/3/4/6/12*) were expressed during the fruit development (Figure 4B). More than 60% of the *AcRFS* genes were significantly induced in response to ABA treatments (Figure 4B). Moreover, *AcRFS10* and *AcRFS11* were significantly up-regulated in response to JA treatments (Figure 4B).

### 2.5. Expression Profiles Analysis of AcGolS and AcRFS Genes in Response to Abiotic and Biotic Stresses

To further explore the potential function of *AcGolS* and *AcRFS* genes under abiotic and biotic stresses, the expression pattern of 9 *AcGolS* and 12 *AcRFS* genes was analyzed through the transcriptome data (Appendix A). As indicated in Figure 5A, *AcGolS4/6/7* exhibited similar expression patterns in response to cold stress between cold-sensitive *A. arguta* variety ‘Kuilv male’ (KL) and cold-tolerant *A. arguta* variety ‘Ruby-3’ (RB), and their expression in RB was significantly higher in KL after cold stress, whereas *AcGolS2/5/8* exhibited no significant change. Under salt stress, *AcGolS2* and *AcGolS4* were up-regulated and peaked at 10d in salt-tolerant *A. deliciosa* variety ‘Guichang’ (GC), and more highly in salt-tolerant than in salt-sensitive *A. chinensis* variety ‘Hongyang’ (HY) (Figure 5A). Transcript levels of *AcGolS1* and *AcGolS6/7* in HY were instantaneously up-regulated at 10d, but these genes in GC were not significantly affected across all the time points (Figure 5A). *AcGolS3/4/8* were significantly induced at 24 h after *Pseudomonas syringae* pv. *actinidiae* (Psa) treatment in resistant *A. eriantha* variety ‘Huate’ (HT), and were significantly higher than susceptible *A. chinensis* variety ‘Hongyang’ (HY) at all-time points (Figure 5A). *AcGolS6* and *AcGolS7* were significantly up-regulated (fold change >2) from 48 h to 96 h in *A. chinensis* variety ‘Hongyang’ (HY), while were relatively low at whole time points in *A. eriantha* variety ‘Huate’ (HT) under Psa treatment (Figure 5A).

*AcRFS11* and *AcRFS12* were down-regulated under cold stress at whole time points in the *A. arguta* variety ‘Ruby-3’ (RB) (Figure 5B). *AcRFS1* and *AcRFS8* were significantly induced at 1h in RB while they were no significant changes in *A. arguta* variety ‘Kuilv male’ (KL) after cold stress treatment (Figure 5B). 80% of the *AcRFS* genes had the highest expression level at 10d in *A. deliciosa* variety ‘Guichang’ (GC) with salt treatment (Figure 5B). *AcRFS1/5/8* were down-regulated under salt stress in both GC and HY varieties. At 24 h, the transcriptional levels of *AcRFS5* and *AcRFS8* in the *A. eriantha* variety ‘Huate’ (HT) were considerably higher than those in the *A. chinensis* variety ‘Hongyang’ (HY) in response to Psa (Figure 5B). The expression levels of *AcRFS11* in HT increased from 0 h to 24 h and then decreased (Figure 5B). The expression of *AcRFS11* was 2-fold lower in HY during the whole Pas treated time than in HT at 24 h (Figure 5B). 

### 2.6. Network Analysis of Hub TF Targeting to Key AcGolS and AcRFS Genes

Co-expression network of hub transcription factors targeting key *AcGolS* and *AcRFS* genes was constructed under cold, salt and Psa stresses. *AcGolS4* under cold treatment, *AcGolS2* under salt treatment and *AcGolS3* under Psa treatment were selected as key *AcGolS* genes and all transcription factors (2000 TFs) for co-expression analysis. *AcRFS4* under cold and salt treatment and *AcRFS7* under Psa treatment were selected as key *AcRFS* gene and all transcription factors (2000 TFs) for co-expression analysis. As shown in Figure 6, the hub *AcNAC*, *AcMYB*, *AcERF*, *AcWRKY*, *AcbHLH* and so on transcription factors related to abiotic and biotic stresses were observed. There was a strong correlation between *AcNAC30* and *AcRFS4* with high expression levels under salt treatment (Figure 6B). Co-expression network analysis indicated that these hub transcription factors genes in the network might play an important role in responding to the abiotic (cold and salt) and biotic stresses (Psa).

### 2.7. Verification of Key AcGolS and AcRFS Genes Expression under Abiotic Stresses

To further validate the expression pattern of *AcGolS* and *AcRFS* genes under abiotic stresses, the stress-responsive *AcGolS* (*AcGolS1*/2/4/8) and *AcRFS* (*AcRFS2/4/8/11*) genes were selected for qRT-PCR analysis. As shown in Figure 7A, *AcGolS1* and *AcGolS2* were significantly induced after NaCl, drought (DT), and waterlogging (WT) stresses. *AcGolS4* showed the highest expression level in response to cold and heat stresses. *AcRFS2/4/11* were significantly up-regulated in response to cold and NaCl treatments. Only *AcRFS8* was strongly expressed under heat and cold treatments. The results confirmed that these genes were really induced by abiotic stresses.

### 2.8. Salt Stress Regulating AcRFS4 Promoter

To further investigate salt-induced *AcRFS4* expression, the 2000 bp promoter sequence of *AcRFS4* was fused to the β-glucuronidase (GUS) coding region and transformed into wild-type *Arabidopsis* (WT). Analysis of three independent transgenic plants (*AcRFS4Pro*::GUS-1, -2, and -3) revealed that salt treatment strongly induced GUS expression in the transgenic lines relative to controls (Figure 8). This finding indicated that the transcription of *AcRFS4* was enhanced by salt stress.

### 2.9. Functional Characterization of AcRFS4 in Response to Salt Stress

To assess the role of *AcRFS4* under salt stress, three transgenic homozygous T3-generation *Arabidopsis* (OE1, OE2 and OE3) were obtained. As shown in Figure 9A, wild-type (WT) and transgenic plants grew well, without obvious difference, under normal conditions (control, CK). Under 150 mmol L^−1^ NaCl, transgenic plants survived but WT were very small and their leaves turned yellow (Figure 9B). Quantitative measurements indicated that the root lengths of transgenic plants were significantly longer than wild-type under salt stress (Figure 9C). These results indicated that *AcRFS4* over-expression indeed increased the tolerance to salt stress in transgenic *Arabidopsis*. Moreover, the raffinose contents in OE plants were significantly higher than that of WT under control condition (Figure 9D), suggesting that *AcRFS4* has a conserved role in raffinose biosynthesis. Under salt stress, OE plants accumulated even more raffinose than WT (Figure 9D), indicating that *AcRFS4* increased the salt tolerance possibly by enhancing raffinose accumulation.

### 2.10. Subcellular Localization of the AcNAC30 Protein

The transient expression of *35S::GFP* plasmid showed that GFP fluorescence was observed throughout the cells, while the green fluorescence signals of the *35S::AcNAC30**-GFP* plasmid were specifically found in the nucleus (Figure 10). This result indicated that *AcNAC30* might encode a nuclear-localized protein, which was consistent with its functional characteristics in regulating gene transcription.

### 2.11. AcNAC30 a Key Transcription Factor Regulating AcRFS4

The transcript level of *AcNAC30* was significantly up-regulated upon salt application (Figure 11A). Compared with control, the transcript level of *AcNAC30* was nearly 25 times higher after salt treatment (Figure 11A). Promoter self-activation of Y1H Gold showed that ≥150 ng mL^−1^ aureobasidin A (AbA) inhibited the growth of Y1H Gold containing the pABAi-*AcRFS4*-Pro recombinant plasmid (Figure 11B). Only the positive control strain and transformed Y1H Gold harboring both pABAi-*AcRFS4*-Pro and pGADT7-*AcNAC30* could grow in a medium without leucine (-Leu) (150 ng mL^−1^ AbA), validating protein-DNA interaction of *AcNAC30* and the *AcRFS4* promoter (Figure 11B). 

## 3. Discussion

Raffinose, the smallest member of RFOs, is widely found in the leaves, roots, seeds and tubers of plants, for instance, *Arabidopsis thaliana* [36], *Brassica napus* [37], *Nicotiana tabacum* [13], *Zea mays* [14], *Cicer arietinum* [38], *Gossypium hirsutum* [39], *Sesamum indicum* [17] and so on. As two significant oligosaccharides in the raffinose family, raffinose and galactinol play a vital role in responding to abiotic and biotic stresses in the plant [4]. Galactitol and raffinose synthase enzymes are critical for raffinose synthesis [5]. To date, there were a few reports regarding *GolS* and *RFS* genes in horticultural crops, such as cucumber [40], peas [41], apple [12], banana [42], tea [7] and grapevine [24]. Kiwifruit is an important horticultural crop with enriching in vitamin C [26]. Here, we identified 22 *GolS* and 24 *RFS* genes from the genome sequence of *A. chinensis* variety ‘Hongyang’ and *A. eriantha* variety ‘White’. As shown in Figure 1A, phylogenetic analysis showed that 21 *GolS* genes were classified into four subgroups (Figure 1A), which is consistent with that previously reported in other species, such as cotton [19], tomato [16], tobacco [13], and sesame [17]. As shown in Figure 1B, phylogenetic analysis showed that 24 *RFS* genes were classified into six subgroups, consistent with cotton [19]. A similar exon/intron structure with 1 to 5 and conserved motifs of most *RFS* gene members shared in the same group was found in kiwifruit, which was also supported by phylogenetic relationships (Figure 2B). In short, these results showed that *RFS* gene family in kiwifruit might be relatively conservative during evolution.

The previous studies showed that gene family expansion and evolution of new functions frequently occurred during gene duplication, particularly in adaptation to abiotic and biotic stresses [43]. The segmental and tandem gene duplications as two significant factors in gene family generation and maintenance may correspond to functional differences among gene family members during gene evolution [44]. In the current study, only one tandem duplication event of *GolS* gene family was found in *A. chinensis*, while it was not found in *A. eriantha* (Appendix A). This result is consistent with that previously reported in Casava [15]. High segmental duplications (more than 80% of genes) were observed in kiwifruit *GolS* and *RFS* gene families (Figure 3), as suggested by previous findings in *Solanum lycopersicum* and *Brachypodium distachyon* [16]. Segmental duplication exhibited exceptionally different expression patterns. There was often a strong link between the function and expression pattern of segmental repeat genes. *AcGolS4/6/7* genes belonging to segmentally duplicated gene pairs in the same subgroup (Figure 1) respond to cold, salt and Psa stresses (Figure 5A). Together, these results indicated that gene duplication might play potential roles in providing genetic sources with novel biological functions during the evolution of the kiwifruit *GolS* and *RFS* gene family.

It has been reported that *GolS* and *RFS* genes may play a key role in synthesizing galactinol or raffinose and regulating the response to abiotic and biotic stresses [4]. There are still no papers investigating *GolS* and *RFS* gene-mediated tolerance to abiotic and biotic stresses in kiwifruit despite extensive researches on other plant species. Comprehensive expression profiles of *AcGolS* and *AcRFS* genes under different hormone and abiotic stresses treatments were analyzed via transcriptome data from this study and previous studies (Appendix A). *AcGolS2* and *AcGolS3* were slightly induced by JA treatments (Figure 4A), which is consistent with the report by Li et [15], Under high concentrations of ABA treatment, galactitol, raffinose contents and galactitol synthase activities were significantly higher than those in control, indicating that exogenous ABA induces the accumulation of RFO in somatic embryos of alfalfa (*Medicago sativa* L.) [45]. In this study, we found that most *AcGolS* and *AcRFS* genes were significantly activated by ABA (Figure 4A), suggesting that ABA signaling may regulate these genes. Seki et al. (2002) suggested that *AtGolS1* and *AtGolS2* slightly induced ABA, while ABA does not induce *AtGolS3* in *A. thaliana* [46]. Promoter analysis of the Chickpea *CaGolS* gene revealed that *CaGolS2* was more sensitive to ABA treatment, which might be related to the ABRE core sequence (ACGT) enriched in its promoter sequence [47]. Under ABA treatment, *ZmVP1* and *ZmABI5* interacted to regulate *ZmGolS2* expression to promote raffinose accumulation in maize seeds [48]. So far, few articles have shown that ABA treatment can promote the expression of *RFS* genes. Notably, the gene expression up-regulation of six *AcRFS* genes (*AcRFS1/4/6/7/10*) was induced transcriptionally in ABA treatment (Figure 4B), suggesting that activation of *AcRFS* gene might be positively correlated with ABA levels. Exogenous ABA-treated improved levels of *VvRafs1* gene expression in grapevine buds [49]. Unsurprisingly, abiotic stresses induced *GolS* and *RFS* genes in plants [50]. Over-expression of *CsGolS1* in cucumber enhanced the assimilate translocation efficiency and accelerated the growth rates of sink leaves, fruits and whole plants under cold stress [51]. The normal growth and physiological processes in kiwifruit were seriously affected under salt stress [28]. Under salt stress, *AcGolS2* and *AcGolS4* up-regulated and peaked at 10d in salt-tolerant *A. deliciosa* variety ‘Guichang’ (GC) and more highly than in salt-sensitive *A. chinensis* variety ‘Hongyang’ (HY) (Figure 5A), which suggested that they were salt-responsive *AcGolS* genes. As above-mentioned, *MeGolS1* reached a rapid peak expression at 12 h in response to salt conditions [15]. *TsGolS2* was up-regulated under salt stress, and overexpression of *TsGolS2* enhanced tolerance to salt in *Arabidopsis* [52]. In addition, overexpression of *PtrGolS3* resulted in higher RFO content and other stress-related metabolites (proline, salicylic acid, amino acids, etc.) compared with wild type, which may increase RFO in woody plants under short-term salt treatment understanding of metabolism [53]. In *Arabidopsis*, *AtRS5* (At5g40390) was reported to participate in abiotic stresses using a reverse genetic approach [22]. *ZmRFS* overexpressing *Arabidopsis* plants displayed a significantly tolerance to drought stress [54]. In this study, the expression levels of *AcRFS2/4/8/11* were higher in abiotic stress-tolerant (cold and salt) kiwifruit cultivars and significantly higher than those in abiotic stress-insensitive kiwifruit cultivars (Figure 5B), which suggests that these *AcRFS* genes might be positively involved in cold and salt tolerances of kiwifruit. A previous study indicated that Na^+^ is toxic for kiwifruit [55]. Therefore, we can’t exclude the possibility that the increased expression of *AcRFS* genes under salt stress might be caused by the toxicity of Na^+^, although there is no report indicating that Na^+^ is toxic to all kiwifruit varieties. Besides, our study found that overexpression of *AcRFS4* in *Arabidopsis* enhanced the salt tolerance of transgenic plants. More raffinose was significantly accumulated in *AcRFS4*-OE *Arabidopsis* lines than in WT under salt stress (Figure 9D). Therefore, we speculate that overexpression of *AcRFS4* would accumulate more raffinose for combating with the salt stress in kiwifruit. So far, a few studies have shown how *GolS* and *RFS* genes are transcriptionally regulated under abiotic and biotic stresses. The *BhWRKY1* transcription factor was involved in dehydration tolerance by regulating *BhGolS1* gene in *Boea hygrometrica* [56]. *ZmDREB1A* directly regulated *ZmRFS* to improve raffinose biosynthesis and enhance plant tolerance to cold stress [57]. Many elements related to phytohormones, biotic and abiotic stresses were observed in the upstream 2000 bp promoter of the *GolS* and *RFS* genes in *A. chinensis* and *A. eriantha* (Appendix A). There was a strong correlation between *AcNAC30* and *AcRFS4* by WGCNA analysis under salt (Figure 6). *AvNAC030* can increase plants’ salt tolerance by improving ROS removal efficiency and maintaining the intracellular and extracellular osmotic balance to protect the integrity of the membrane [58]. By qRT-PCR analysis, it was shown that the transcription of both *AcNAC30* and *AcRFS4* was significantly induced under salt stress (Figure 7B and Figure 11A). Apart from this, GUS staining result of *AcRFS4* promoter revealed that the transcription of *AcRFS4* was induced by salt stress (Figure 8). Considering the results of Y1H, we believe that the transcription factor *AcNAC30* may activate *AcRFS4* expression by directly binding its promoter (Figure 11B). This work provides a foundation for future investigation of *AcGolS*, *AcRFS* and *AcNAC30* gene functions for kiwifruit’s abiotic and biotic stresses tolerance.

## 4. Materials and Methods

### 4.1. Identification and Phylogenetic Analysis of GolS and RFS Genes

The *A. chinensis* variety ‘Hongyang’ v3.0 and *A. eriantha* variety ’White’ genome sequences were obtained from the Kiwifruit Genome Database (http://kiwifruitgenome.org/ (accessed on 4 May 2021)) [59]. First, the BLASTP tool was used to screen *AtGolS* and *AtRFS* genes in *Arabidopsis* against all of the protein sequences in each kiwifruit genome, using an E-value threshold of 1.0. Secondly, the profile hidden Markov model (PF05691 and PF01501) of the Pfam 32.0 database (http://pfam.xfam.org/ (accessed on 2 June 2021)) was further applied to confirm the hits under an E-value <1.0 [60]. We removed the genes, which did not include conserved domains of *GolS* and *RFS* genes. Finally, these remaining candidate protein sequences were used to calculate physicochemical properties by the online tool ProtParam, including molecular weight (MW), theoretical isoelectric point (pI), and hydrophilic mean (GRAVY). The possible subcellular location of *GolS* and *RFS* genes in kiwifruit was predicted using the WolfPSORT tool [61]. Multiple protein sequences were aligned by using MUSCLE [62]. The Neighbor-Joining (NJ) phylogenetic trees were constructed using MEGA software with 1000 bootstrap replicates [63]. *AtGolS* and *AtRFS* amino acid sequences were downloaded from TAIR (https://www.arabidopsis.org/ (accessed on 2 August 2021)) in this study.

### 4.2. Analysis of Gene Structure, Protein Motif and Cis-Element in Promoter Regions 

The gene structures were presented using the online Gene Structure Display Service (http://gsds.cbi.pku.edu.cn (accessed on 10 August 2021)) [64]. The conserved motifs were displayed by using the online tool MEME Suite5.1.1 (version 5.1.1; http://meme-suite.org/ (accessed on 16 August 2021)) [65]. The gene promoters from the initiation codon (2000 bp before ATG) were retrieved from the kiwifruit genome. Then, the putative cis-regulatory elements in the promoter region sequences were predicted via the PLACE database (http://www.dna.affrc.go.jp/PLACE/ (accessed on 30 August 2021)). 

### 4.3. Analysis of Gene Duplication, Synteny and Chromosomal Locations

The potential gene duplication and events synteny relationships between kiwifruit and *A. thaliana* of *GolS* and *RFS* gene families were determined using MSCanX software [66]. The physical location maps of chromosomes were drawn using MapInspect 1.0 software.

### 4.4. Expression Analysis of AcGolS and AcRFS Genes 

The RNAseq raw sequence data of 78 samples covering diverse tissues at different developmental time points were downloaded from the NCBI website (Bioproject ID PRJNA324539) [67]. The raw transcriptome data of two *A. arguta* genotypes, ‘Kuilv male’ (KL) and ‘Ruby-3’ (RB), with high and low freezing tolerance, respectively, at −25 °C for 0 h, 1 h, and 4 h were obtained from the NCBI database, with project number PRJNA248163 [68]. We also downloaded the RNA sequencing raw data of resistant *A. eriantha* variety ‘Huate’ and susceptible *A. chinensis* variety ‘Hongyang’ at 0, 12, 24, 48, and 96 h after inoculation with Psa from the NCBI (Bioproject ID PRJNA514180) [31]. The rest of the transcriptome raw data (unpublished) came from our own research group. Trimmomatic was performed to filter the raw sequence datas [69]. The clean datas were adopted to map the reference *A. chinensis* variety ‘Hongyang’ v3.0 genome using HISAT2 [70]. The genes were quantified with the featureCounts package in R. The heatmaps were constructed using the TBtools software [71].

### 4.5. Co-Expression Network Analysis of Hub Transcription Factors

Co-expression network hub transcription factors targeting key *AcRFS* and *AcGolS* genes were analyzed by using the ‘cor’ function of the R package with Pearson’s correlation coefficients (r ≥ 0.9 or r ≤ −0.9 and *p*-values ≤ 0.05) under cold, salt and Psa stresses conditions. Cytoscape 3.4.0 software (http://www.cytoscape.org/ (accessed on 30 August 2021)) was used to visually describe co-expression network results [72]. The degree >N50 was used to identify the genes of hub transcription factors.

### 4.6. Plant Material and Abiotic Stresses Treatments

*A. chinensis* variety ‘Hongyang’ (HY) tissue culture seedlings were transferred to a soil mixture of perlite and sand (3:1, *v*/*v*). All seedlings were grown in a growth chamber at a temperature of 18 °C (night) and 24 °C (day), relative humidity of 60–80%, and a 14/10 h photoperiod (daytime, 06:00–20:00). The seedlings were irrigated with water once every two days. After two months, they were randomly divided into six groups for stresses treatments. For heat and cold stress treatment, the seedlings were transferred into two chambers with the temperature set at 48 °C and 4 °C, respectively [73]. Treated seedlings were harvested at 6 h after treatment. For salt, drought and waterlogging stresses, the seedlings were soaked in 0.6% NaCl for 6 days [28]. the seedlings were flooded for 7 days [74], and the seedlings were dried for 14 days [75]. Non-treated seedlings were used as the control (CK). All samples were immediately frozen in liquid nitrogen and stored at −80 °C. 

### 4.7. RNA Isolation and Quantitative Real-Time PCR (qRT-PCR) Analysis 

According to the manufacturer’s instructions, total RNA was extracted from all samples using the RNAprep pure Plant Kit (TIANGEN, Beijing, China). For qRT-PCR analysis, first-strand cDNAs were synthesized from DNaseI-treated total RNA using the Hifair® III 1st Strand cDNA Synthesis kit (Yeasen, Shanghai, China) according to the manufacturer’s instructions. qRT-PCR was performed on the Biorad CFX96 real-time PCR system using the ChamQ SYBR qPCR Master Mix (Vazyme, Nanjing, China). The relative expression levels were calculated by using 2^−∆∆Ct^ method. The qRT-PCR assays were performed with three biological and technical replicates. The gene-specific primers are listed in Appendix A.

### 4.8. Isolation of the AcNAC30 and AcRFS4 Full-Length CDS, and AcRFS4 Promoter Sequence 

The CTAB method was implemented to extract the total DNA of *A. chinensis* variety ‘Hongyang’ (HY)leaves [76]. Transcription factor *AcNAC30* was identified based on the sequence in the kiwifruit database (http://kiwifruitgenome.org/ (accessed on 6 October 2021), Accession Nos. Actinidia09980). *AcNAC30* and *AcRFS4* full-length CDS were cloned from reverse-transcribed cDNA with specific primers (Appendix A). The *AcRFS4* promoter was cloned from extracted DNA with *AcRFS4*-Pro primers (Appendix A). The cloning conditions were as follows: predenaturation for 3 min at 98 °C; 30 cycles of template denaturation for 10 s at 98 °C, primer annealing for the 60 s at 58 °C, and 30 s at 72 °C; and a final extension step of an additional 5 min at 72 °C. The amplified products were connected to the pESI-T vector and transferred into DH5α Chemically Competent Cell, and the recombinant plasmids were obtained through white spot screening.

### 4.9. Analysis of AcRFS4 GUS Histochemical Staining and AcNAC30 Protein Subcellular Localization Assay 

The 20,000 bp *AcRFS4* promoter sequence was inserted into pCAMBIA1301 to drive the GUS reporter gene by using BglII and BamHI. *AcRFS4Pro::GUS* transgenic *Arabidopsis* of 10-day-old seedlings in MS/2 media. Then, transgenic *Arabidopsis* were treated with 150 mM NaCl for 24 h. Finally, transgenic *Arabidopsis* were used for histochemical staining. GUS staining was performed according to Jefferson et al. [77]. Primers for construction of the *AcRFS4Pro*::GUS vector were provided in Appendix A.

The CDS of *AcNAC30* without the stop codon was constructed into the vector pCAMBIA1305-35S-GFP vector using double enzyme digestion (cut with Xbal and BamHI). Then, the plasmid (*35S::AcNAC30-GFP*) was transferred into Agrobacterium strain EHA105. This strain was injected into 1-month-old tobacco leaves. Finally, the GFP fluorescence was detected by confocal laser scanning microscopy. Primers used to construct the *35S::AcNAC30-GFP* vector were provided in Appendix A.

### 4.10. Transcription Activation Assay in Yeast

The yeast one-hybrid (Y1H) assays were conducted based on the method previously described [78]. The AcRFS4 promoter sequence containing three core cis-elements (CATGT binding site) was inserted into a pAbAi vector via the HindII and SalI sites (pABAi-AcRFS4-Pro primers in Appendix A). The recombinant vector of pABAi-AcRFS4-Pro was linearized with BstBI and transferred into Y1H Gold through PEG/LiAc. The full-length region of AcNAC30 was embedded into the pGADT7 vector (AD) through EcoRI and SalI sites (pGADT7-AcNAC30 primers in Appendix A). Transformed Y1H Gold harboring pABAi-AcRFS4-Pro and pGADT7-AcNAC30 were cultured to test the interaction on SD/-Leu with aureobasidin A (150 ng mL^−1^ AbA) for three days at 30 °C. Finally, an autoactivation analysis was conducted according to the manufacturer’s protocol. The p53-promoter fragment and pGADT7-Rec (AD-Rec-P53) co-transform Y1H Gold as a positive control. Empty pGADT7 was transformed in Y1H Gold with pABAi-AcRFS4-Pro as a negative control.

### 4.11. Plasmid Construction, Transgenic Arabidopsis Generation and Salt Stress Tolerance 

Complete CDS of AcRFS4 was cloned between the NcoI and BstEII sites of the pCAMBIA1302 vector with the In-Fusion and a pair of primers (Appendix A). The AcRFS4-1302 vector was initially transformed to Agrobacterium strain GV3101. Finally, it was transformed to Arabidopsis using the floral dip method [79]. The AcRFS4-1302 transgenic Arabidopsis of T1, T2, and T3 generations was selected on 1/2 MS medium with 25 mg·L^−1^ Hygromycin B (HYG). T3 homozygous of these transgenic lines were used for salt stress tolerance assays. The seeds of WT and homozygous T3 transgenic Arabidopsis (OE1, OE2 and OE3) were separately placed on 1/2 MS medium containing 150 mmol·L^−1^ NaCl. After 7 days, the root lengths of WT and transgenic Arabidopsis was measured by using Image J 1.8.0. The leaves of WT and transgenic Arabidopsis under salt stress treatments were collected, then raffinose were extracted from WT and transgenic Arabidopsis leaves according to Li et al. [80]. Raffinose was determined by high performance liquid chromatography (HPLC) with amide column (4.6 mm × 150 mm id., 5 μm;). A Waters X-bridge amide column (Waters, USA) was washed by methanol/H_2_O (90:10) as the mobile phase at speed of 0.5 mL/min for separation of soluble sugar components. An evaporative light-scattering detector (ELSD; Waters 2424) was applied to monitor the sugar signal.

## 5. Conclusions 

The galactinol synthase (*GolS*) and raffinose synthetase (*RFS*) are two critical enzymes synthesizing raffinose, which play an important role in modulating plant growth and a variety of biotic or abiotic stresses. A total of 22 *GolS* and 24 *RFS* genes were identified in *A. chinensis* and *A. eriantha*, respectively. We comprehensively analyzed the *GolS* and *RFS* gene families and their involvement in kiwifruit’s abiotic and biotic stresses responses at the genome-wide scale. RNA-seq and qRT-PCR analyses revealed that *AcGolS1* and *AcGolS2* were significantly induced after NaCl, DT, and WT stresses. *AcRFS4* was significantly up-regulated in response to NaCl treatments. The GUS staining result revealed that the transcriptional regulation of *AcRFS4* was influenced by salt stress. Overexpression of *AcRFS4* in *Arabidopsis* proved that this gene enhanced the salt tolerance of transgenic plants at physiologic (higher raffinose content). A yeast one-hybrid assay demonstrated that *AcNAC30* could interact with the *AcRFS4* promoter. Therefore, we speculate that *AcNAC30* promotes the expression of *AcRFS4* gene to increase the accumulation of raffinose, thereby improving the salt stress tolerance of kiwifruit.

## Figures and Tables

**Figure 1 ijms-23-08836-f001:**
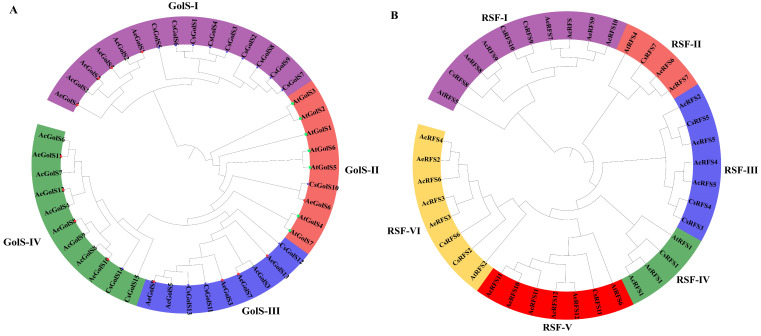
Phylogenetic analysis of kiwifruit and other species *GolS* and *RFS* proteins. The phylogenetic trees were conducted based on the full-length amino acid sequences using MEGA 6.0 by the neighbor-joining method with 1000 bootstrap replicates. (**A**) The deduced full length amino acid sequences of *A. chinensis* (*AcGolS*), *A. eriantha* (*AeGolS*), *C. sinensis* (*CsGolS*) and *A. thaliana* (*AcGolS*) were used for phylogenetic tree construction. (**B**) The deduced full length amino acid sequences of *A. chinensis* (*AcRFS*), *A. eriantha* (*AeRFS*), *C. sinensis* (*CsRFS*) and *A. thaliana* (*AcRFS*) were used for phylogenetic tree construction.

**Figure 2 ijms-23-08836-f002:**
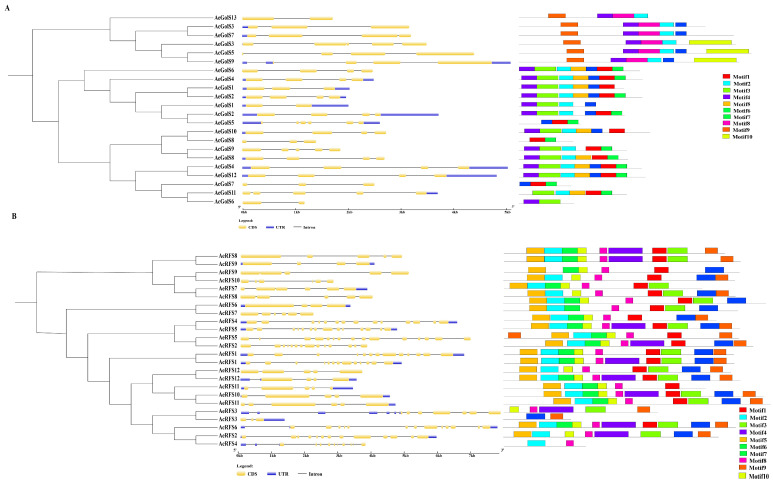
Conserved motifs and exon-intron organization of *GolS* and *RFS* genes in two kiwifruit species. (**A**) Neighbour-joining phylogenetic tree, gene structures, and conserved motifs of the *GolS* genes in *A. chinensis* and *A. eriantha*. (**B**) Neighbour-joining phylogenetic tree, gene structures, and conserved motifs of the *RFS* genes in *A. chinensis* and *A. eriantha*.

**Figure 3 ijms-23-08836-f003:**
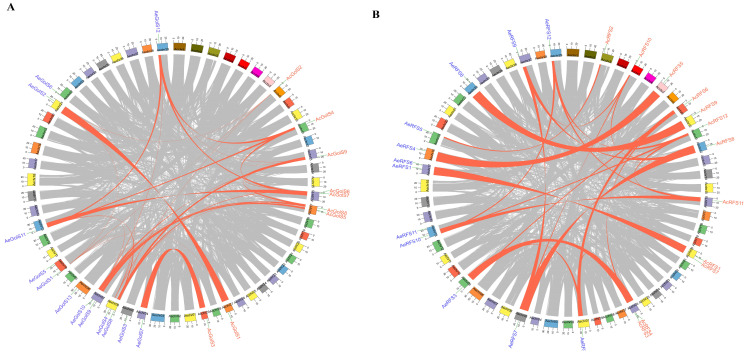
Gene duplication analysis of the kiwifruit *GolS* and *RFS* genes. (**A**) Pink lines connect the syntenic regions between the kiwifruit *GolS* genes. (**B**) Pink lines connect the syntenic regions between the kiwifruit *RFS* genes.

**Figure 4 ijms-23-08836-f004:**
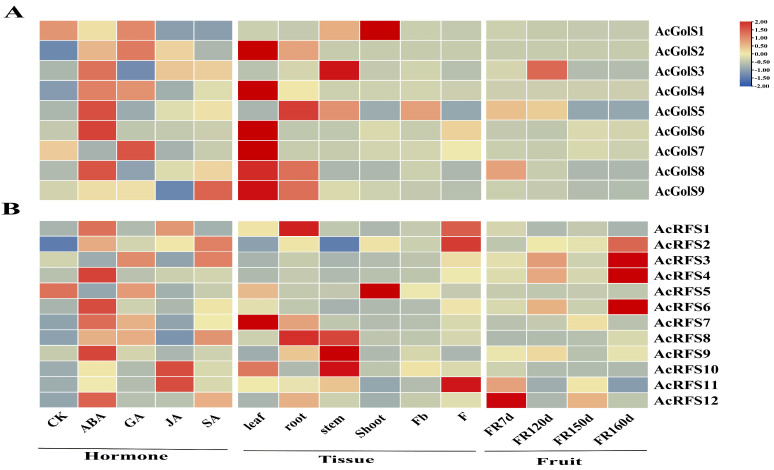
Hormone-induced conditions, tissue-specific and fruit development expression analysis of *AcGolS* and *AcRFS* genes in *A. chinensis.* (**A**) Expression profiles of *AcGolS* genes; (**B**) Expression profiles of *AcRFS* genes. The color bar to the right of the figures represents the log2 FPKM (fragments per kilobase of exon per million reads mapped) value.

**Figure 5 ijms-23-08836-f005:**
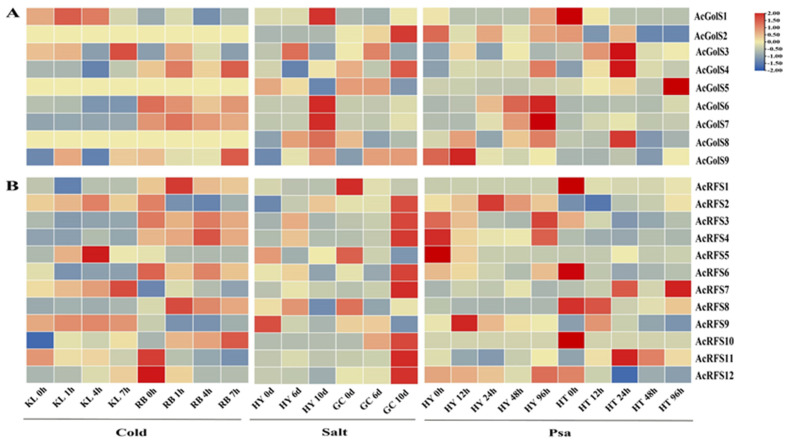
Expression analysis of *AcGolS* and *AcRFS* genes under cold, salt, and Psa. (**A**) Expression profiles of *AcGolS* genes; (**B**) Expression profiles of *AcRFS* genes. To the figures’ right, the color bar represents the log2 FPKM (fragments per kilobase of exon per million reads mapped) value.

**Figure 6 ijms-23-08836-f006:**
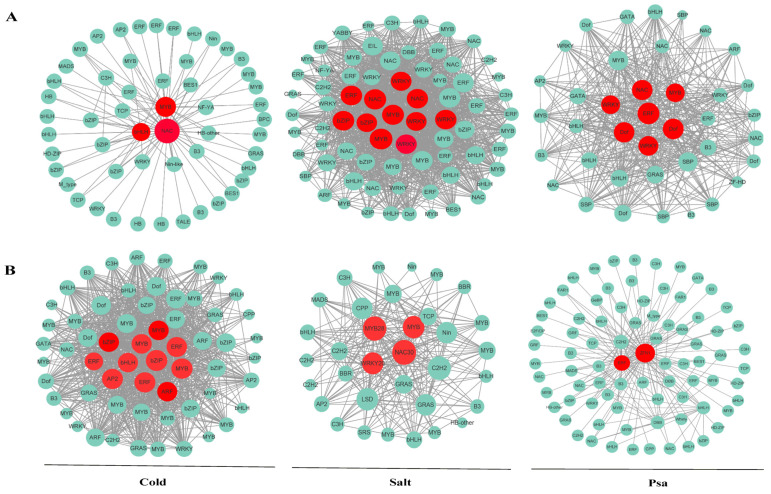
Network maps of hub TF targeting key *AcGolS* (**A**) and *AcRFS* (**B**) genes under cold, salt and Psa stresses. An adjusted network map of a subset of the module genes interacts with the hub TF genes (weight of >0.5).

**Figure 7 ijms-23-08836-f007:**
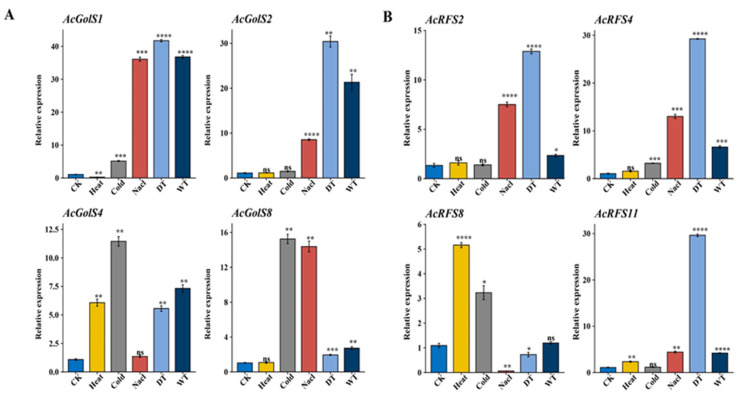
Expression levels of key *AcGolS* and *AcRFS* genes under abiotic stresses (heat, cold, salt, drought, waterlogging) by quantitative real-time RT-PCR. (**A**) Expression levels of *AcGloS1/2/4/8*. (**B**) Expression levels of *AcRFS2/4/8/11*. Three biological and technical replicates calculated the error bars. Asterisks indicated the corresponding gene significantly up or down-regulated under the different treatments using *t*-test (* *p* < 0.05, ** *p* < 0.01, *** *p* < 0.001, **** *p* < 0.0001; one-way ANOVA test by Tukey’s test). Ns indicated the corresponding gene no significantly up or down-regulated under the different treatments using *t*-test.

**Figure 8 ijms-23-08836-f008:**
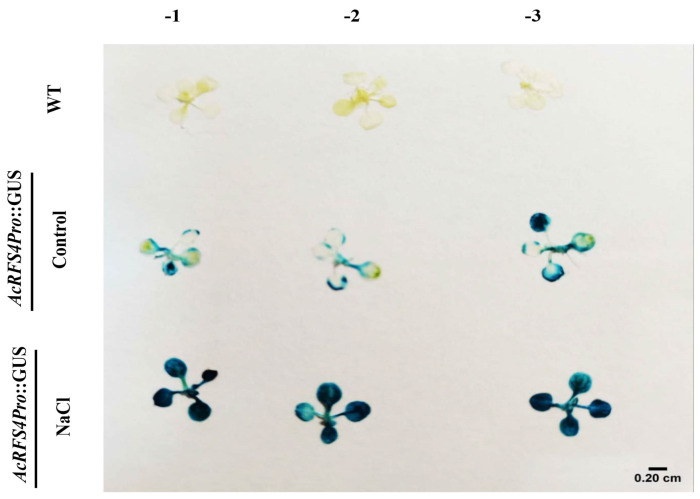
*AcRFS4Pro*::GUS transgenic *Arabidopsis* of 10-day-old seedlings grown in MS/2 media without (Control) and with 150 mM NaCl for 24 h. Markers -1, -2 and -3 indicate three independent lines of wild type (WT) and transgenic plants (*AcRFS4*Pro::GUS). Scale bars = 0.2 cm.

**Figure 9 ijms-23-08836-f009:**
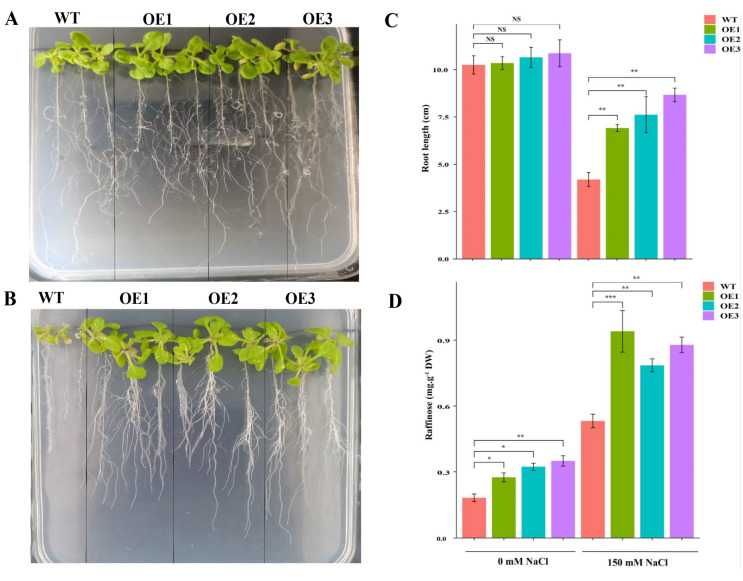
Overexpression of *AcRFS4* results in enhanced salt tolerance. (**A**) Observation of the root length of wild type (WT) and T3 *AcRFS4*-transgenic *Arabidopsis* seedlings (OE1, OE2 and OE3) under 0 mM NaCl. (**B**) Observation of the root length of wild type (WT) and T3 *AcRFS4*-transgenic *Arabidopsis* seedlings (OE1, OE2 and OE3) under 150 mM NaCl. (**C**) Measurement of the root length of wild type (WT) and T3 *AcRFS4*-transgenic *Arabidopsis* seedlings (OE1, OE2 and OE3) under different concentrations of NaCl. (**D**) Raffinose content of wild type (WT) and T3 *AcRFS4*-transgenic *Arabidopsis* seedlings (OE1, OE2 and OE3) under different concentrations of NaCl. Three biological replicates and technical replicates calculated the error bars. Asterisk indicates significant difference compared to WT (* *p* < 0.05, ** *p* < 0.01, *** *p* < 0.001; one-way ANOVA test by Tukey’s test). NS indicates no significant difference compared to WT.

**Figure 10 ijms-23-08836-f010:**
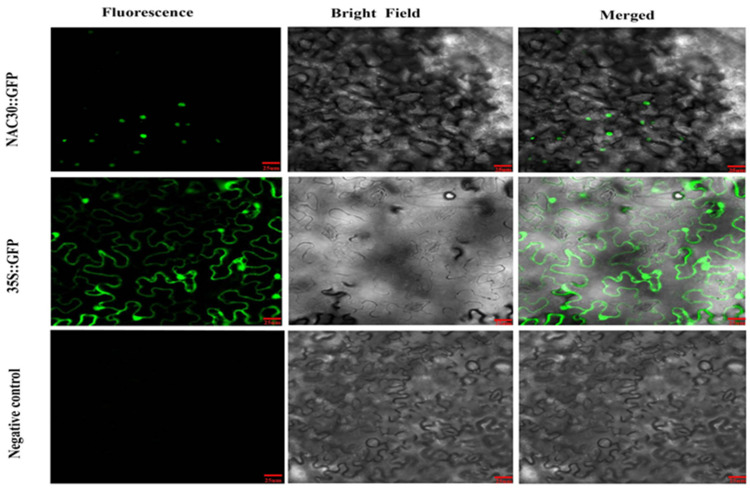
Subcellular localization of *AcNAC30* and empty vector in *Nicotiana benthamiana* leaves after 2 days of infiltration. The green fluorescence of GFP was indicated. White arrows indicated the GFP signal from the nucleus. Scale bars = 25 μm.

**Figure 11 ijms-23-08836-f011:**
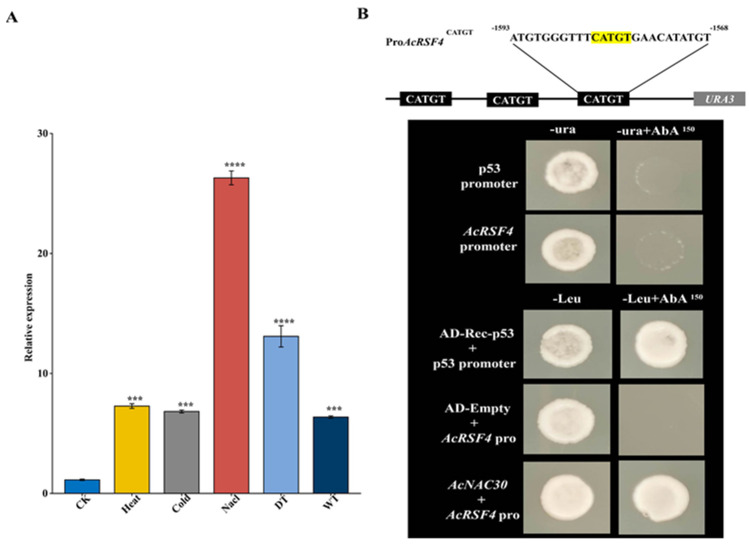
Identification of *AcNAC30* transcription factor modulating *AcRFS4* in kiwifruit. (**A**) Expressions analysis of *AcNAC30* under abiotic stresses. (**B**) Interaction of *AcNAC30* with the promoter of *AcRFS4* in the Y1H assay. Three biological replicates calculated the error bars. Asterisks indicated the corresponding gene significantly up-or down-regulated under the different treatments using *t*-test (*** *p* < 0.001, **** *p* < 0.0001).

## Data Availability

The datasets generated for this study can be found in SRA, the accession number: PRJNA691387, PRJNA681641 and PRJNA514180.

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
