# Peer review of "Genome-Wide Expression Profiling Analysis of Kiwifruit GolS and RFS Genes and Identification of AcRFS4 Function in Raffinose Accumulation"

_ijms, 2022, doi:10.3390/ijms23168836_

Round 1

Reviewer 1 Report

Summary

The paper is a report of the profiling of genome wide gene expression of genes involved in Raffinose metabolism of kiwifruit. The knowledge reported is of interest for understanding expression of raffinose metabolism pathway genes in kiwifruit in response to various abiotic and biotic stresses. It is a combination of analysis of RNA-seq data from other publications and targeted RNA expression from an abiotic stress experiment carried out. The conclusion of the paper is that there is a correlation between AcNAC30 and AcRSF4 which in itself lacks substance since the authors then speculate that this leads to accumulation of raffinose helping in salt stress tolerance.

Concept comments

I found it very hard to distinguish what was carried out by the authors in respect to their experiments in contrast to data analysed from experiments which has been previously published. I also note that the papers where the RNA-seq data was sourced are not in the paper references. The sources of the results are interwoven. It is extremely difficult to understand how the information was attained and why. In the experiment carried out by the authors NaCl was used as the treatment for salt stress. Yet it has been reported that Na is toxic for kiwifruit and other work carried out on assessing kiwifruit responses to salt stress use KCl (Klages,K et al Accumulation of myo-inositol in Actinidia seedling subjected to salt stress. Annals of Botany v84 Issue 4 pp521-527). The abbreviation for the raffinose synthase is set as RFS, yet the universal abbreviation is RS. I would suggest that RS is used as there are numerous places in the text where the raffinose synthase is referred to as RSF not RFS. The speculation that upregulation of the gene expression means raffinose is being synthesised as an osmoprotectant needs confirmation by increases in the metabolite content. In fact, if this was part of the paper it would make a far greater impact and prove the concept. Overall, the paper lacks cohesiveness and a clear proven outcome.

Specific comments

All paper RSF instead of RFS

Line 23  AcGloS1/2/4/8 change to AcGolS1/2/4/8  

Line 41  Space between many and compatible

Line 42  Space between many and soluble

Line 54  In what species was the over expression?

Line 59  State it was in Arabidopsis

Line 64 Make all plants names in lower case

Line 65  Species name should be in italics

Line 70  Arabidopsis thalina name should be in italics

Line 78  Actinidia should be in italics

Line 84 The reference used reports only on drought and dormancy

Line 92  Space between of and our

Line 98  A. thaliana in italics

Line 110                Please clarify what is mean in the sentence beginning “Besides, “

Line110 Change AeGols to AeGolS

Figure 1.               Why are they GolSs and RSs proteins instead of GolS and RS proteins

Line 133                What does RAR stand for, it is not noted anywhere earlier in the paper.

Line 164               Change RFS and GolS to Gols and RFS as the rest of the paper talks about them in this order.

Line 188, 189, 194, 214 and many more examples             ..”salicylic acid treatments” This refers to the experiment in another paper and would be clearer if the paper was referenced. “only AcGolS9 was strongly expressed etc”. The refers to a treatment made in the paper where the authors obtained the RNA data. This should be referenced.

Line 274                What do DT and WT stand for?

Line 276               NaCl

Line 282                A: “ ….” then no b: instead Scheme 2. ??

Line 295-304       This needs expanding. There is no mention that the tobacco was transformed etc and then Figure 9 says its tobacco.

Figure 10.            What do the asterix on the bars denote?

Line 319                in kiwifruit. What variety? Since the paper talks about so many different sources of information and species this needs to be reported.

Line 323                First word should read Raffinose.

Line 331                Change one of the most important to an important.

Line 334-35         Sentence starting “A. eriantha “ makes it sound like the vitamin C content makes it Psa tolerant which is not what the paper says.

Line 342                Put HY and HT in words as the abbreviations don’t confer instant recognition and it makes it hard to follow.

Line 362                Here you use the botanical name of Casava and Line 326 you use Casava. Consistency.

Line 398                Kiwi fruit is a vine not a tree.

Line 534-542       Papers where RNA-seq data is obtained aren’t referenced.

Line 548-549       No refence to papers where work was carried out so there is no record of what the treatments were.

Line 560, 561, 563 and 564           Alphabetical groups noted but they are nowhere else in the paper

Line 659                Details of RNA-seq data used in this study yet these publications aren’t even referenced!

Reviewer 2 Report

1.       Be sure the gene name and latin word, enzyme name in italic, ; Arabidopsis thaliana, Arabidopsis and A. thaliana need to be unified in the whole text;

2.       The abstract need to be cut down;

3.       Figures quality were poor;

4.       “data from published and RNA-seq data”, this should have reference citation;

5.       Results 2.4 and 2.5, just the expression data presented, where is the key point, what key information for readers;

6.       Line 269” Quantitative Reverse 269 Transcription PCR”, should be deleted.

7.       Figure 8, the display of the treatment and repeat are confusing;

8.       How the author introduced NAC30 transcription factor;

9.       Delete Figure 11, as there is no strong evident to support it;

10.   Check the type writing carefully, line 609-610;  

Round 2

Reviewer 1 Report

MAjor flaws have not been addressed 

Reviewer 2 Report

The manuscript was revised as reviewer requirements, which can be published after carefully language checking.

Round 3

Reviewer 1 Report

This paper has a small portion of original experimental work which is hidden amongst other researchers results. Referencing the works where the results were obtained from other works within the results would make it clearer and easier to see how the results were obtained as the fact that the RNA is from other researchers work is not clearly defined within the text. The whole background of how the data was obtained is not clear in the paper itself. It is structured poorly and the story is muddled and confusing. For a journal of this impact factor the paper is not up to scratch with its story and the evidence of the conclusions. Reference 19 with a similar report for sesame includes metabolite analyses of the tissues where the gene expression has been attained. 

The experimental work within the paper is poorly designed and uses a salt which is toxic to kiwifruit to elicit a salinity response when in fact the response being elicited could be more catastrophic in nature and not a water related response.

Though there is an increase in expression of raffinose synthase related genes again there is no proof that raffinose is being synthesised as the response to the treatments as there are no quantifications included within the paper. The authors response to this is that they have evidence in arabidopsis. Where is the evidence in kiwifruit as these two plants have different carbohydrate profiles and metabolisms as kiwifruit synthesises planteose. Trisaccharides similar to raffinose have been reported to be synthesised using RFS so the authors cannot conclude that raffinose is involved without proof of the compound itself being elevated within kiwifruit. The paper by Klages et el which was in the first response to the authors as a point of reference to salt stress in kiwifruit does NOT report elevation of raffinose and in contrast reports elevation of myo-inositol which is in involved in the synthesis of raffinose with GolS. This work was carried out prior to the discovery of the presence of planteose in kiwifruit.

There are still errors with RFS written in diagrams and text as RSF. If the authors had used RS as suggested this would have been easier to check for consistencies.
